# Enhancing Healthcare Model Trustworthiness through Theoretically Guaranteed One-Hidden-Layer CNN Purification

**Hanxiao Lu**[1,2]*, **Zeyu Huang**[1,3]*, **Ren Wang**[1]†
[1]Illinois Institute of Technology  [2]Columbia University  [3]Beijing Normal University

## Abstract

The use of Convolutional Neural Networks (CNNs) has brought significant benefits to the healthcare industry, enabling the successful execution of challenging tasks such as disease diagnosis and drug discovery. However, CNNs are vulnerable to various types of noise and attacks, including transmission noise, noisy mediums, truncated operations, and intentional poisoning attacks. To address these challenges, this paper proposes a robust recovery method that removes noise from potentially contaminated CNNs and offers an exact recovery guarantee for one-hidden-layer non-overlapping CNNs with the rectified linear unit (ReLU) activation function. The proposed method can recover both the weights and biases of the CNNs precisely, given some mild assumptions and an overparameterization setting. Our experimental results on synthetic data and the Wisconsin Diagnostic Breast Cancer (WDBC) dataset validate the efficacy of the proposed method. Additionally, we extend the method to eliminate poisoning attacks and demonstrate that it can be used as a defense strategy against malicious model poisoning.

## 1 Introduction

Health care is coming to a new era where abundant medical images are playing important roles. In this context, Convolution Neural Networks (CNNs) have captured tremendous attention in handling a surge of biomedical data because of their efficiency and accuracy. Overcoming medical challenges is becoming possible with the help of CNNs. Examples of the CNN model's application include detecting tuberculosis in chest X-ray images (Liu et al., 2017), diagnosing COVID-19 through chest X-ray image classification (Reshi et al., 2021), and predicting abnormal health conditions using unstructured medical health records (Ismail et al., 2020). Experimental results have shown that CNN model could achieve as high as $99.5\%$ accuracy in terms of COVID-19 disease detection (Reshi et al., 2021). However, CNN models are usually delivered or trained in untrusted environments and can be easily contaminated (Gündüz et al., 2019; Ma et al., 2022). The performance of polluted CNN could be greatly reduced and thus its credibility is weakened, which brings about severe medical accidents such as clinical misdiagnosis and treatment failure. Thus, an efficient model purification algorithm is needed to maintain a reliable CNN model in the field of medical care. Few studies have explored how to purify fully-connected neural networks to reduce the negative impact of unexpected noise from a robust recovery perspective (Gao & Lafferty, 2020) and Bayesian estimation (Shao et al., 2021). In this paper, we for the first time consider the recovery of a one-hidden-layer CNN polluted by some noises from an arbitrary distribution. We further extend the proposed recovery method to detoxify CNNs under training-phase poisoning attacks (Gu et al., 2019; Bai et al., 2023).

**Our contributions.**   By properly selecting design matrices in the proposed robust recovery method, all CNN parameters can be purified to ground-truth parameters, as demonstrated in this work.

---

*The first two authors contributed equally to this paper. This work was done when Hanxiao Lu and Zeyu Huang were research interns at the Trustworthy and Intelligent Machine Learning Research Group at the Illinois Institute of Technology.

†This work was supported by the National Science Foundation (NSF) under Grant 2246157. Correspondence to Ren Wang (rwang74@iit.edu)

Additionally, we establish a quantitative relationship between learning accuracy and noise level. Synthetic and Wisconsin Diagnostic Breast Cancer (WDBC) data experiments confirm the theoretical correctness and method effectiveness, in addition to the novel contributions to the theoretical analysis of CNNs. Furthermore, we leverage the proposed method to purify CNNs trained on poisoned data, which differs from previous works that focused on detection and fine-tuning. Our approach aims to directly remove the noisy weights corresponding to the poisoning effect, and only requires a small amount of clean data from resources other than the training set.

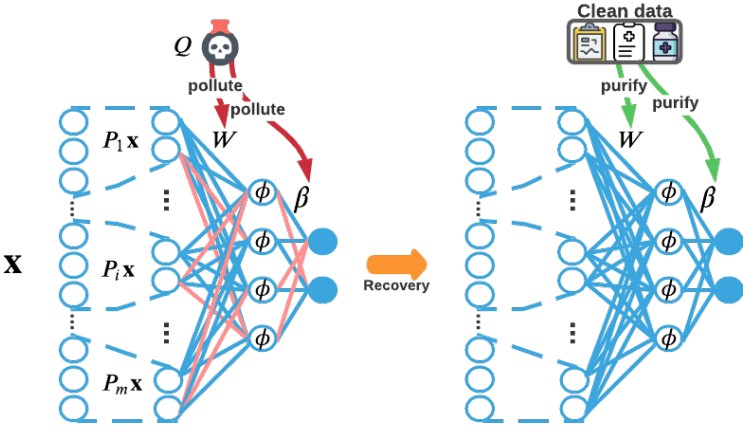

Figure 1: **The proposed convolutional neural network (CNN) purification method aims to remove noises from contaminated weights.** Hidden-layer weights $W$ and output layer weights $\beta$ of a CNN may be contaminated by noises or corruptions. The proposed framework can directly recover $W$ and $\beta$ with clean data points.

## 2    PROBLEM FORMULATION

In this section, we begin by providing an overview of the problem of purifying CNNs. We then proceed to describe the CNN architecture and the contamination model that we investigate in this study. Specifically, we consider the scenario where a CNN is trained on a set of $n$ inputs $\{\mathbf{x}_s\}_{s=1}^n \in \mathbb{R}^d$ along with their corresponding ground truth labels $y_s$. The network's parameters are assumed to be contaminated by random noise $z$, which is independent of the input data and is generated from an arbitrary distribution. Such noise can arise from either post-training phase perturbations or poisoned inputs. Our primary objective is to purify the contaminated CNN parameters using a proposed robust recovery method that avoids the need to retrain the model from scratch.

## 3    CNN AND CORRUPTION MODEL

As illustrated in Figure 1, this work studies the one-hidden-layer CNN architecture:

$$\hat{y}_s = \sum_{j=1}^p \sum_{i=1}^m \beta_j \psi(W_j^T P_i \mathbf{x}_s) \ , \tag{3.1}$$

where $\mathbf{x}_s \in \mathbb{R}^d$ is the input and the scalar $\hat{y}_s$ is its prediction. Following the same setting as in previous theoretical works on CNNs (Zhong et al., 2017), we consider CNN with $m$ non-overlapping input patches. $P_i \mathbf{x_s} \in \mathbb{R}^k$ is the $i$-th patch ($i = 1, 2, \cdots, m$) of input $\mathbf{x_s}$, which is separated by $m$ matrices $\{P_i\}_{i=1}^m \in \mathbb{R}^{k \times d}$ defined as follows.

$$P_i = [\underbrace{\mathbf{0}_{k \times k(i-1)}}_{\text{All zero matrix}} \quad \underbrace{I_k}_{\text{Identity matrix} \in \mathbb{R}^{k \times k}} \quad \underbrace{\mathbf{0}_{k \times k(m-i)}}_{\text{All zero matrix}}]$$

Note that the non-overlapping setting forces $\{P_i\}_{i=1}^m$ independent of each other and therefore simplifies our proofs. $W = [W_1, W_2, \cdots, W_p] \in \mathbb{R}^{k \times p}$ denotes the hidden layer weights with each column $W_j \in \mathbb{R}^k$ representing the $j$-th kernel weights. The Rectified Linear Unit (ReLU) operation

$\psi$ is the most commonly used activation function that transforms data $t$ into $\text{ReLU}(\cdot) = \max(0, \cdot)$. $\beta \in \mathbb{R}^p$ denotes the output layer weights and $\beta_j$ is its $j$-th entry. In this paper, we consider an overparameterization setting, where $p, k \gg n$.

Here we define the contamination model for $W$ and $\beta$.

$$\Theta_j = W_j + z_{W_j} \quad , \tag{3.2}$$

$$\eta = \beta + z_\beta \quad , \tag{3.3}$$

where $\Theta$ and $\eta$ are contaminated parameters of CNN's hidden layer and output layer, respectively. The vectors $z_{W_j} \in \mathbb{R}^k, z_\beta \in \mathbb{R}^p$ are noise vectors with each entry $[z_{W_j}]_i$ ($[z_\beta]_i$) generated from an arbitrary distribution $Q_i$ with fixed probability $\epsilon$, which is between 0 and 1. In the post-training phase poisoning scenario, our contamination model describes the additional noises added to clean weights $W$ and $\beta$. In the training phase poisoning scenario we considered in this work, additional noises are injected through manipulated training data. According to recent research (Wang et al., 2020; Pal et al., 2023), some CNN weights contain a portion of poisoning information, which our contamination model can also characterize.

## 4 PURIFICATION OF ONE-HIDDEN-LAYER CNN ALGORITHM

**CNN model training.** Before introducing the CNN recovery optimization and algorithm, we need to specify the process of obtaining $W$ and $\beta$. In our setting, the one-hidden-layer CNN is trained by the traditional gradient descent algorithm, which is shown in Algorithm 1. $X \in \mathbb{R}^{d \times n}$ is the matrix format of the training examples. $W(0), \beta(0)$ are initializations of hidden and output layers' weights. They are initialized randomly following Gaussian distributions $\mathcal{N}(0, k^{-1}I_k)$ and $\mathcal{N}(0, 1)$, respectively. $\gamma$ and $\frac{\gamma}{k}$ are learning rates indicating step sizes of gradient descents. With the purpose of easier computation of the partial derivative of loss function $\mathcal{L}$ with respect to $\beta$ and $W$, we use the squared error empirical risk

$$\mathcal{L}(\beta, W) = \frac{1}{2} \frac{1}{n} \sum_{s=1}^{n} (y_s - \frac{1}{\sqrt{p}} \sum_{j=1}^{p} \sum_{i=1}^{m} \beta_j \psi(W_j^T P_i \mathbf{x_s}))^2$$

that quantifies the prediction errors of the learned CNN. $\frac{1}{\sqrt{p}}$ is used for simplifying our proofs. Note that in the post-training phase poisoning scenario, $W(t_{max})$ and $\beta(t_{max})$ are the ground truth we want to extract from observations $\Theta$ and $\eta$. We will introduce the details of the training phase poisoning scenario in Section 6. We use the following $\ell_1$ norm-based robust recovery optimization method to achieve accurate estimations.

---

**Algorithm 1** CNN Model Training

---

   **Input:** Data $(y, X)$, maximum number of iterations $t_{max}$
   **Output:** $W(t_{max})$ and $\beta(t_{max})$
   Initialize $W_j(0) \sim \mathcal{N}(0, k^{-1}I_k)$ and $\beta_j(0) \sim \mathcal{N}(0, 1)$ independently for all $j \in [p]$.
   **for** $t = 0$ **to** $t_{max}$ **do**
      **for** $j = 1$ **to** $p$ **do**
         $\beta_j(t) = \beta_j(t-1) - \gamma \frac{\partial \mathcal{L}(\beta(t-1), W(t-1))}{\partial \beta_j(t-1)}$
      **end for**
      **for** $j = 1$ **to** $p$ **do**
         $W_j(t) = W_j(t-1) - \frac{\gamma}{k} \frac{\partial \mathcal{L}(\beta(t), W(t-1))}{\partial W_j(t-1)}$
      **end for**
   **end for**
   **Output:** $\beta(t_{max})$ and $W(t_{max})$

---

**Robust recovery for CNN purification.** The $\ell_1$ norm-based recovery optimizations for $W$ and $\beta$ are defined as

$$\widetilde{u}_j = \arg\min_u \|\Theta_j - W_j(0) - A_W^T u_j\|_1 \quad , \tag{4.1}$$

$$\widetilde{v} = \arg\min_{v} \|\eta - \beta(0) - A_\beta^T v\|_1 \quad , \tag{4.2}$$

where $A_W$ is the design matrix for purifying $W$:

$$A_W = [P_1 X, P_2 X ..., P_m X] \quad , \tag{4.3}$$

$A_\beta$ is the design matrix for recovering $\beta$:

$$A_\beta = \left[ \sum_{i=1}^m \psi(W^T P_i x_1), ..., \sum_{i=1}^m \psi(W^T P_i x_n) \right] \quad , \tag{4.4}$$

$\widetilde{u}_j, \widetilde{v}$ are the optimal estimations of the models' coefficients of the two optimization problems. The key for successful recovery of $W_j$ out of $\Theta_j$ is that $W_j(t_{max}) - W_j(0)$ lies in the subspaces spanned by $A_W$. Similarly, we can recover $\beta$ out of $\eta$ because $\beta_j(t_{max}) - \beta_j(0)$ lies in the subspace spanned by $A_\beta$. Further conditions which are necessary for successful recovery of $W_j, \beta$ are theoretically analyzed in theorem 5.2. Based on equation 4.1 and equation 4.2, the purification of contaminated one-hidden-layer CNN is given in Algorithm 2. By properly selecting the design matrix of the hidden layer recovery $A_W$ and the design matrix of the output layer $A_\beta$, one can make a successful recovery.

---

**Algorithm 2** Purification of One-hidden-Layer CNN

---

**Input:** Contaminated model $(\eta, \Theta)$, design matrix $A_W, A_\beta$, and parameter initialization $\beta(0), W(0)$.
**Output:** The repaired parameters $\widetilde{\beta}$ and $\widetilde{W}$
**for** $j = 1$ **to** $p$ **do**
$\quad \widetilde{u}_j = \arg\min_{u} \|\Theta_j - W_j(0) - A_W^T u_j\|_1$
$\quad \widetilde{W}_j = W_j(0) + A_W^T \widetilde{u}_j$
**end for**
$\widetilde{v} = \arg\min_{v} \|\eta - \beta(0) - A_\beta^T v\|_1$
$\widetilde{\beta} = \beta(0) + A_\beta^T \widetilde{v}$
**Output:** $\widetilde{W}$ and $\widetilde{\beta}$

---

**Design Matrix of hidden layer** $A_W$. We now explain in detail why we choose $A_W$ in the format of equation 4.3. We define the mapping from input to output as $f(\mathbf{x_s}) = \frac{1}{\sqrt{p}} \sum_{j=1}^p \sum_{i=1}^m \beta_j \psi(W_j^T P_i \mathbf{x_s})$. For weights update in each iteration of the Algorithm 1, the partial derivative of the loss function with respect to $W_j$ is represented by $\frac{\partial \mathcal{L}(\beta, W)}{\partial W_j}\Big|_{(\beta, W) = (\beta(t), W(t-1))} = \sum_{s=1}^n \sum_{i=1}^m \alpha_i P_i \mathbf{x_s}$ where $\alpha_i$ sums up all other remaining terms .

One can easily observe that the gradient $\frac{\partial \mathcal{L}(\beta, W)}{\partial W_j}$ lies in the subspace spanned by $P_i \mathbf{x_s}$. And this indidates that vector $W_j(t_{max}) - W_j(0)$ also lies in the same subspace. Therefore, we can use the design matrix $A_W$ in the format of equation 4.3 to purify CNNs' weights.

**Design Matrix of output layer** $A_\beta$. We then introduce how we select $A_\beta$ in the form of equation 4.4 and how it helps the recovery. For weights update in each iteration of the Algorithm 1, the partial derivative of the loss function with respect to $\beta$ is $\frac{\partial \mathcal{L}(\beta, W)}{\partial \beta_j}\Big|_{(\beta, W) = (\beta(t-1), W(t-1))} = \sum_{s=1}^n \delta_s \sum_{i=1}^m \psi(W_j^T(t-1) P_i \mathbf{x_s})$ where $\delta_s$ sum ups all other remaining terms.

Since the derivative of $\mathcal{L}$ with respect to the $j$-th entry $\beta_j$ is represented by combinations of $\sum_{i=1}^m \psi(W_j^T(t-1) P_i \mathbf{x_s})$ and $\delta_s$ only depends on $\mathbf{x_s}$, we get the conclusion that $\frac{\partial \mathcal{L}(\beta, W)}{\partial \beta}$ lies in the subspace that is spanned by $\sum_{i=1}^m \psi(W^T(t-1) P_i \mathbf{x_s})$. Further notice that $\beta(t_{max}) - \beta(0)$ is an accumulation of $\frac{\partial \mathcal{L}(\beta, W)}{\partial \beta}$ in each iteration. Unlike the subspace spanned by $P_i \mathbf{x_s}$ which is used for hidden layer recovery remains constant, the subspace spanned by $\sum_{i=1}^m \psi(W^T(t-1) P_i \mathbf{x_s})$ which is used for output layer recovery keeps changing over $t$. However, thanks to overparametrization

assumption of CNN, one could show $W(t)$ obtained by Algorithm 1 is close to initialization $W(0)$ for all $t \geq 0$. Theorem 5.1 in the next section shows that $W(t)$s are all not far away from each other. Thus, $\beta(t_{max}) - \beta(0)$ approximately lies in the same spanned subspace, resulting in the proposed design matrix $A_\beta$.

## 5 THEORETICAL RECOVERY GUARANTEE

In the previous section, we introduced our CNN purification algorithm and went over how to build design matrices for recovering the hidden and output layers. In this section, we demonstrate theoretically that the proposed algorithm's estimation is accurate. We assume $\mathbf{x}_s$ follows Gaussian distribution $\mathcal{N}(0, I_d)$ for $\forall s \in [n]$ with $|y_s| \leq 1$. Let $f_s(t)$ be $f(\mathbf{x_s})$ with weights $W_j(t)$ and $\beta_j(t)$. We then have the following conclusion.

**Theorem 5.1.** *If* $\frac{mnlog(mn)}{k}$, $\frac{(mn)^3log(p)^4}{p}$ *and* $mn\gamma$ *are all sufficiently small, then*

$$\max_{1 \leq j \leq p}||W_j(t) - W_j(0)|| \leq \frac{100mnlog(p)}{\sqrt{pk}} = R_W \tag{5.1}$$

$$\max_{1 \leq j \leq p}||\beta_j(t) - \beta_j(0)|| \leq 32\sqrt{\frac{(mn)^2log(p)}{p}} = R_\beta \tag{5.2}$$

$$||y - f_s(t)||^2 \leq (1 - \frac{\gamma}{8})||y - f_s(0)||^2 \tag{5.3}$$

*for all* $t \geq 1$ *with high probability.*

Although weights $W(t)$ and $\beta(t)$ are updated over iterations $t$, Theorem 5.1 tells us that the post-trained parameter $W$ and $\beta$ via Algorithm 1 are not too far away from their initializations. Due to the bounded distance, we can show that $\beta(t_{max}) - \beta(0)$ approximately lies in the subspace spanned by $A_\beta$. Moreover, the distance between the ground truth $y$ and the final prediction is bounded by the distance between $y$ and the model's initial prediction, indicating a global convergence of Algorithm 1 despite the nonconvexity of the loss.

Assisting by Theorem 5.1, we propose the main theorem below to demonstrate that Algorithm 2 can effectively purify CNN. Under Algorithm 1, the following conclusion holds.

**Theorem 5.2.** *Under condition of theorem 5.1 with additional assumption that* $\frac{log(p)}{k}$ *and* $\epsilon\sqrt{mn}$ *are sufficiently small, then* $\widetilde{W} = W(t_{max})$ *and* $\frac{1}{p}||\widetilde{\beta} - \beta(t_{max})||^2 \lesssim \frac{(mn)^3log(p)}{p}$ *with high probability, where* $W(t_{max})$ *and* $\beta(t_{max})$ *are obtained by gradient descent algorithm and* $\widetilde{W}$ *and* $\widetilde{\beta}$ *are results of purification of CNN.*

According to theorem 5.2 pre-condition $\frac{mnlog(mn)}{k}$, $\frac{(mn)^3log(p)^4}{p}$ and $mn\gamma$, successful model repair requires large number of hidden layer neurons $p$, large partition dimension $k$, small number of partition $m$, small training examples $n$ and small poison ratio $\epsilon$. The assumption $\frac{log(p)}{k}$ further puts the constraint on the distance between $log(p)$ and $k$ in terms of successful parameter repairing. Compared with theorem B.2 in Gao & Lafferty (2020), the constraint contains extra $m$ and replaces $d$ with $k$. Extra $m$ appears since the construction of both design matrices takes account of $m$. And input dimension to feed into CNN is $k$ rather than $d$ of DNN. The reason $\beta$ could not be exactly recovered and has error bound $\frac{(mn)^3log(p)}{p}$ is subspace spanned by $\sum_{i=1}^m \psi(W^T(t-1)P_i\mathbf{x}_s)$ keeps changing over $t$, which has been discussed in section 4. Thus $\beta(t_{max}) - \beta(0)$ approximately lies in the subspace spanned by $A_\beta$.

## 6 EXPERIMENTAL RESULTS

In this section, we conduct experiments on synthetic data and Wisconsin Diagnostic Breast Cancer (WDBC) dataset (Dua & Graff, 2017) to demonstrate the effectiveness of our proposed CNN purification method and evaluate the alignments of the results with our theoretical analysis. Furthermore, experiments show the proposed purification algorithm can be utilized to mitigate poisoning attacks

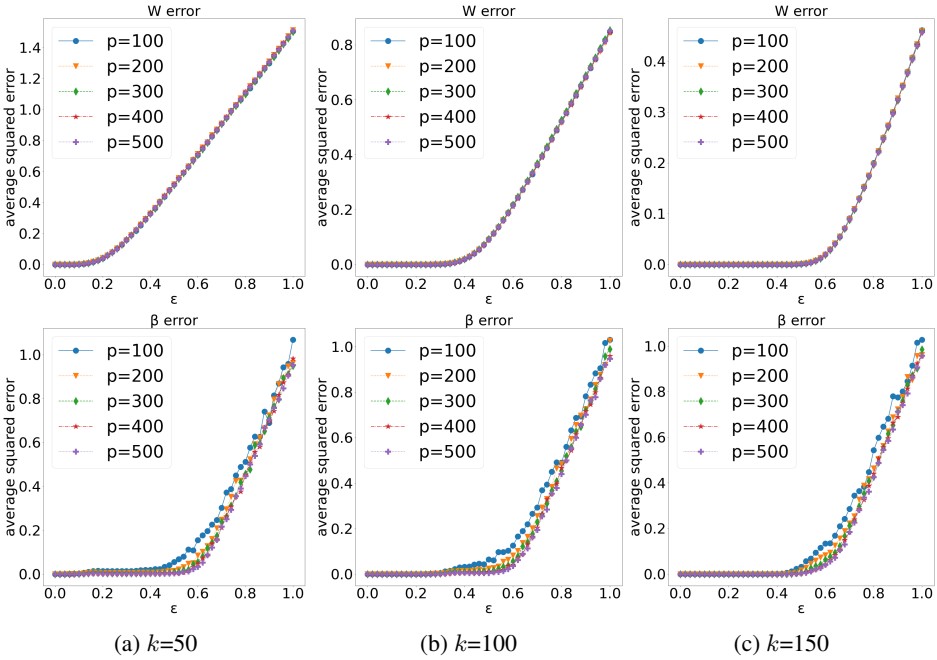

Figure 2: **Increasing $p$ and $k$ promotes the recovery performance on the synthetic dataset** ($n = 5, m = 5$). When $p$ increases, the limit of $\epsilon$ for successful recovery of $\beta$ also increases. When $k$ increases, the limit of $\epsilon$ for successful recovery of $W$ increases.

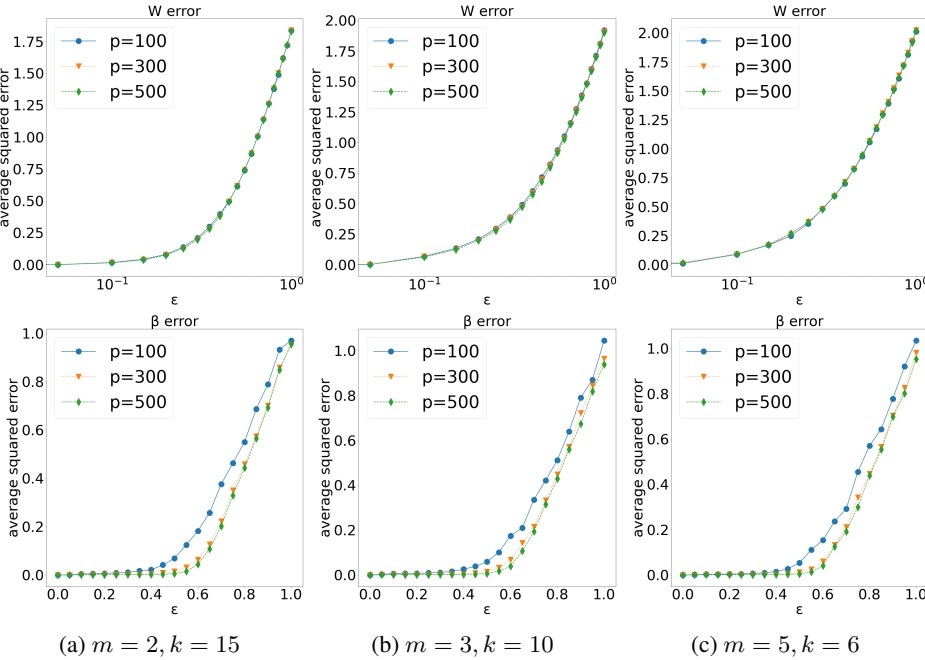

Figure 3: **Increasing $p$ promotes the recovery performance on the WDBC dataset** ($n = 5$). When $p$ increases, the limit of $\epsilon$ for successful recovery of $\beta$ also increases.

from the poisoned CNNs. The error is measured by the average $\ell_2$ error. All the experimental results of synthetic data are averaged over 100 trials. All the experimental results of WDBC data are averaged over 10 trials.

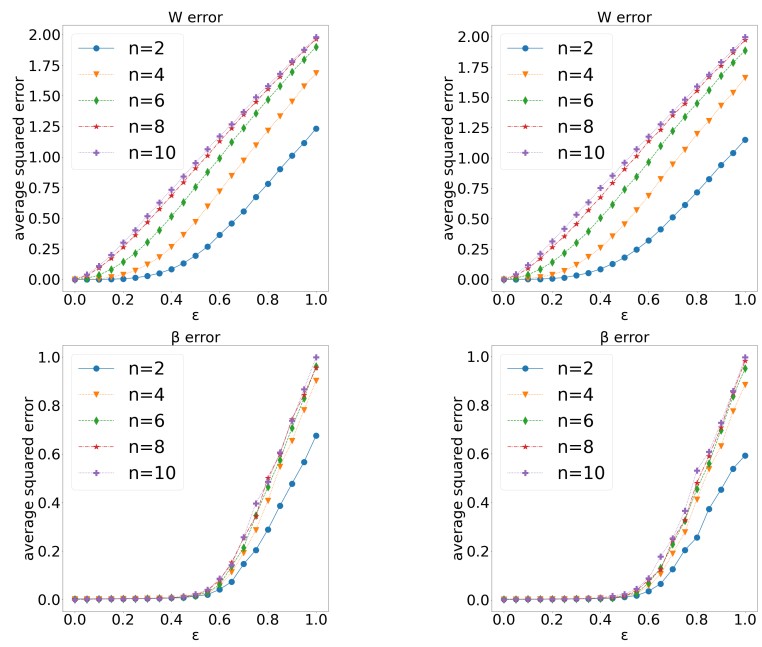

(a) CNN purification by training instances    (b) Model repair by non-training instances

Figure 4: **Our CNN purification method has the ability to yield good performance even when using a limited number of clean data points, which may not necessarily originate from the training dataset (*training batch size* $= 10$).** When $n$ decreases, the limit of $\epsilon$ for successful recovery of both $W$ and $\beta$ also increases

## 6.1 EXPERIMENTS ON SYNTHETIC DATA

The synthetic data are generated by $x_s \sim \mathcal{N}(0, I_d)$. The noises $[z_{W_j}]_i$, $([z_\beta]_i)$ are generated from $\mathcal{N}(1, 1)$. We evaluate $p$ and $k$ by fixing the number of data points $n = 5$ and the number of partitions $m = 5$. Figure 2 shows results of recovery errors under different $p$ with $k = 50, 100, 200$. When $\epsilon$ is small, e.g., $\epsilon < 0.2$, the recovery of both $W$ and $\beta$ are more likely to be successful. In each column, one can see that increasing $p$ further increases the limit of $\epsilon$ for successful recovery of $\beta$. The phenomenon is consistent with Theorem 5.2 as we require $\frac{(mn)^3 log(p)}{p}$ to be small. Across the three columns of figures, an obvious observation is that the limit of $\epsilon$ for successful recovery of $W$ increases when $k$ increases. In our theorems, successful recovery needs $\frac{log(p)}{k}$ and $\frac{mnlog(mn)}{k}$ to be sufficiently small.

## 6.2 EXPERIMENTS ON REAL DATASET

The studied data are randomly selected from the WDBC training dataset, which is a widely used benchmark dataset in machine learning and medical research. It contains features derived from digitized images of fine needle aspirate (FNA) of breast mass and corresponding diagnosis of malignant or benign tumors. The noise $[z_{W_j}]_i$, $([z_\beta]_i)$ are generated from $\mathcal{N}(1, 1)$.

First, we evaluate $p$ by fixing the number of data points $n = 5$. Figure 3 shows results of recovery errors under different $p, m, k$. In each column, one can see that increasing $p$ increases the limit of $\epsilon$ for successful recovery of $\beta$. The phenomenon is similar to that shown in the synthetic data experiment Figure 2 and the same reason applies here.

Then we evaluate the number of data points $n$ used in recovery by fixing the training batch size to be 10. Left column figures in Figure 4 show results of recovery errors by $n$ data points selected from the training batch. Right column figures in Figure 4 show results of recovery errors by $n$ data points selected outside of the training batch. One can see that our CNN purification method can achieve good performance even recovering with a small number of clean data points and potentially not

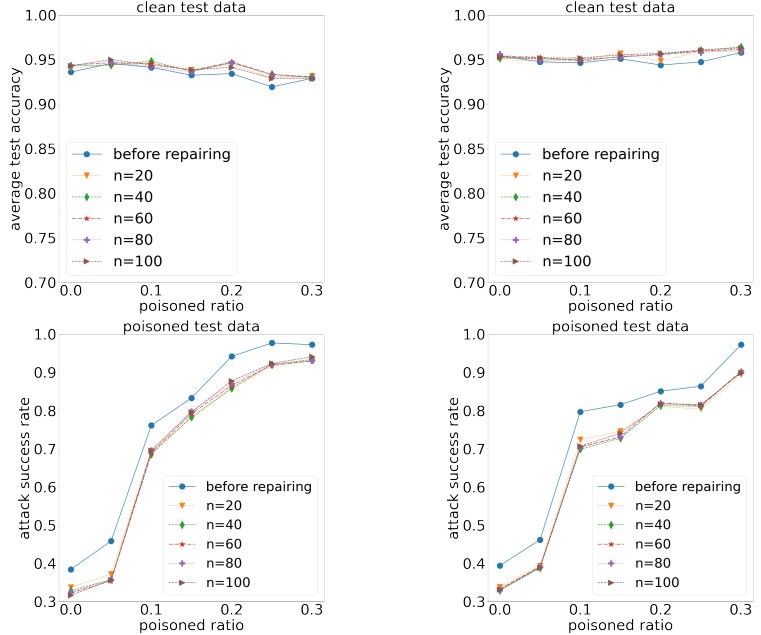

(a) Mitigating poisoning attack by training batch instances (b) Mitigating poisoning attack by instances not in training batch

Figure 5: **Even with a small number of clean data points, CNN purification can mitigate the poisoning effect (*training batch size* $= 100$).** The poisoned ratio indicates the percentage of the poisoned training data. The attack success rate is the percentage of test data that has been successively attacked.

from the training data. The phenomenon is consistent with Theorem 5.2 as we require $\frac{(mn)^3 log(p)}{p}$, $\frac{mnlog(mn)}{k}$ and $\epsilon\sqrt{mn}$ to be small.

## 6.3 POISONING ATTACK MITIGATION

Here we use the same data selected from the WDBC dataset. The training loss is set to cross-entropy loss, which is commonly used in poisoning settings. Noise vectors are poisoned parameters generated from poisoning attacks Gu et al. (2019); Wang et al. (2020). The poisoning attack used in this work aims to force CNNs to predict a target class when the input is injected by a fixed pattern. When the first five features of inputs are set to 5, the outputs of the CNN model will always be 0 (benign). We vary the ratio of poisoned inputs $\epsilon$ by fixing the training batch size to 100. Figure 5 shows results of test accuracy and attack success rate under different $n$. One can see that CNN purification can maintain high average test accuracy and mitigate the poisoning effect even with a small number of clean data points.

## 7 CONCLUSION

CNNs are susceptible to various types of noise and attacks in applications to healthcare. To address these challenges, this paper proposes a robust recovery method that removes noise from potentially contaminated CNNs, offering an exact recovery guarantee for one-hidden-layer non-overlapping CNNs with the rectified linear unit (ReLU) activation function. The proposed method can precisely recover both the weights and biases of the CNNs, given some mild assumptions and an overparameterization setting. We have successfully validated our method on the Wisconsin Diagnostic Breast Cancer (WDBC) dataset. *We emphasize that this work mainly focuses on theoretical analysis. Our future directions are (1) Extending CNN purification to larger healthcare models and large medical datasets (2) Improving the proposed method to eliminate various poisoning attacks on medical models.*

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
