# OpenReview forum: "Enhancing Healthcare Model Trustworthiness through Theoretically Guaranteed One-Hidden-Layer CNN Purification"
_ICLR.cc/2023/Workshop/TML4H — ICLR 2023 Workshop TML4H Poster_

### Official Review · Reviewer_hDSG · 2023-02-24
**Interesting idea to denoise the weights for contaminated one-hidden-layer CNN but the experiment setting is weak**

**Rating:** 6
**Confidence:** 3

**Review:**

## summary of the paper

This paper used a denoising strategy to purify the contaminated one-hidden-layer CNN, which doesn't need to retrain the model. The method is validated on synthetic and breast cancer dataset.

## strength
- The idea of denoising the weights is interesting. This process is data-efficient that does not need the whole training set.

- The proposed algorithm has theoretical recovery guarantee.


## weaknesses
- The title should be specific (e.g., healthcare model -> one-hidden-layer CNN) because only the one-hidden-layer CNN is studied.

- In real healthcare scenarios, the one-hidden-layer CNN is not a common choice. It would be great if the author can demonstrate the effectiveness of the algorithm for commonly used CNN models, eg.g, ResNet

- The experiment setting is weak. It would be great validation the method on a wide range of real medical datasets. Here are some potential candidates
https://grand-challenge.org/challenges/

---

### Official Review · Reviewer_wCxy · 2023-03-01
**I'm unfamiliar with the CNN purification research field and can't accurately judge the work. I have some concerns about the paper, such as the simplicity of the CNN used in the experiments and the many prior constraints for the method. These concerns lead me to question the technique's applicability to other complex deep learning models and its ability to enhance healthcare model trustworthiness.**

**Rating:** 5
**Confidence:** 2

**Review:**

I am not an expert in the field of CNN purification and lack familiarity with this area, making it difficult to make a precise judgment. I understand the purpose of this article. The authors propose a model correction method to correct the "contaminated" weights of the model by using the few gold standard training data. This is an interesting research area, but my concerns are: 1) the convolutional neural network used in the experiment is too simple, only contains a layer of a convolutional network and activation function, and 2) too many prior constraints in the method, such as the assumption of non-overlapping convolutional kernels. Based on these two points, my concern is that the method can not be widely applied to complex neural networks. 3) Only one medical data WDBC was used in the experiments, and the overall method of the paper does not reflect the "ENHANCING HEALTHCARE MODEL TRUSTWORTHINESS THROUGH CNN PURIFICATION" as stated in the title to address the issue of trustworthiness in healthcare. Overall, I feel that the methodology is not very clear, and the experiments are not particularly convincing. Still, I am not a researcher in this field, so I cannot firmly adhere to my judgment above.

---

### Meta-Review · Area_Chair_sHDe · 2023-03-05

**Recommendation:** Accept (Poster)
**Confidence:** 4

**Metareview:**

The paper proposes a recovery method to purify the contaminated one-hidden-layer CNN. Experiments on two datasets, i.e., synthetic data and Breast Cancer dataset are conducted. The reviewers generally appreciate the interesting idea proposed in this work. However, the reviewers have a few concerns. Please address these, especially the questions about the technique's applicability to other complex deep learning models, and generalization on real medical datasets.